# *In vitro* anti-*Helicobacter pylori* activity and *antivirulence activity of* cetylpyridinium chloride

**Mingjin Xun**[1,2], **Zhong Feng**[1,2,3]*, **Hui Li**[1,2], **Meicun Yao**[3], **Haibo Wang**[1,2], **Ruixia Wei**[1], **Junwei Jia**[1,2], **Zimao Fan**[1,2], **Xiaoyan Shi**[1,2], **Zhanzhu Lv**[1,2], **Guimin Zhang**[1,2]*

1 National Engineering and Technology Research Center of Chirality Pharmaceutical, Lunan Pharmaceutical Group Co., Ltd., Linyi, Shandong, China, 2 International Pharmaceutical Engineering Laboratory in Shandong Province, Shandong New Time Pharmaceutical Co., Ltd., Linyi, Shandong, China, 3 School of Pharmaceutical Sciences (Shenzhen), Sun Yat-sen University, Shenzhen, Guangdong, China

* 13734352772@163.com (ZF); lunanzhangguimin@163.com (GZ)

**Data Availability Statement:** All relevant data are within the manuscript.

**Funding:** The author(s) received no specific funding for this work.

## Abstract

The primary treatment method for eradicating *Helicobacter pylori* (*H. pylori*) infection involves the use of antibiotic-based therapies. Due to the growing antibiotic resistance of *H. pylori*, there has been a surge of interest in exploring alternative therapies. Cetylpyridinium chloride (CPC) is a water-soluble and nonvolatile quaternary ammonium compound with exceptional broad-spectrum antibacterial properties. To date, there is no documented or described specific antibacterial action of CPC against *H. pylori*. Therefore, this study aimed to explore the *in vitro* activity of CPC against *H. pylori* and its potential antibacterial mechanism. CPC exhibited significant in vitro activity against *H. pylori*, with MICs ranging from 0.16 to 0.62 µg/mL and MBCs ranging from 0.31 to 1.24 µg/mL. CPC could result in morphological and physiological modifications in *H. pylori*, leading to the suppression of virulence and adherence genes expression, including *flaA*, *flaB*, *babB*, *alpA*, *alpB*, *ureE*, and *ureF*, and inhibition of urease activity. CPC has demonstrated in vitro activity against *H. pylori* by inhibiting its growth, inducing damage to the bacterial structure, reducing virulence and adherence factors expression, and inhibiting urease activity.

## Introduction

*Helicobacter pylori* (*H. pylori*) is a Gram-negative, spiral-shaped, microaerophilic bacterium species that has adapted well to human conditions [1–3]. In a comprehensive study and meta-analysis of 73 countries worldwide, Zamani et al. emphasized that *H. pylori* infect over 44 percent of the world's population, including more than one-half of the inhabitants of developing countries and more than one-third of the inhabitants of developed countries [4]. Infection with *H. pylori* has been linked to the etiology of chronic gastritis, peptic ulcers in the stomach, MALT-lymphoma, and possibly gastric cancer [5–7]. The World Health Organization (WHO) has identified *H. pylori* as one of the most dangerous carcinogens, and gastric cancer is the world's second leading cause of cancer-related death [1,8]. Currently, the two most often used therapies for *H. pylori* infection are bismuth quadruple therapy (using a bismuth on the

**Competing interests:** The authors have declared that no competing interests exist.

foundation of the strategy) and antibiotic-based triple therapy, which consists of two antibiotics and a proton pump inhibitor [1,9]. However, rising antibiotic resistance in *H. pylori* has generated serious concerns regarding antibiotic-based treatment. Antibiotic overuse has enhanced *H. pylori* resistance to routinely used antibiotics [10], posing a significant problem in the management of *H. pylori* infection. A systematic review of *H. pylori* resistance to antibiotics rates, for example, indicated that both the primary and tertiary resistance rates of clarithromycin, metronidazole, and levofloxacin exceeded 15% in 65 countries, calling into question the efficacy of present tripled and quadrupled therapy [11]. In 2017, the World Health Organization (WHO) designated antibiotic-resistant *H. pylori* as a bacterium of high priority, posing a significant threat to human health [12]. Therefore, the development of novel anti- *H. pylori* drugs is of paramount importance.

Cetylpyridinium chloride (CPC) is a water-soluble, nonvolatile quaternary ammonium compound known for its remarkable broad-spectrum antibacterial characteristics [13,14]. It has been used in oral hygiene products for decades to effectively manage oral infections and control supragingival buildup [15–17]. CPC has recently been studied in dental applications, and a novel endodontic sealant incorporating a large dose of CPC showed for a long time antibacterial effects versus *E. faecalis* [18,19]. CPC has been reported to have a killing effect on enveloped viruses [20], including SARS-CoV-2 [21]. Nevertheless, limited research has been conducted on the efficacy of CPC against *H. pylori*, particularly concerning its activity against drug-resistant *H. pylori* strains, and the underlying anti-*H. pylori* mechanism of CPC remains obscure.

In this study, we conducted a methodical investigative analysis of CPC's in vitro antibacterial capacity, putative antibacterial action mechanism, and anti-*H. pylori* potential.

## Materials and methods

### Chemicals and reagents

CPC was purchased from Shaoxing Minsheng Pharmaceutical Co., Ltd. (Shaoxing, China). Amoxicillin (AMO), acetohydroxamic acid (AHA), and clarithromycin (CLR) were obtained from Macklin Biochemical Co., Ltd. (Shanghai, China). Metronidazole (MTZ) was purchased from Sigma-Aldrich LLC. (St. Louis, MO, USA). Levofloxacin (LEF) was purchased from Target Molecule Corp. (Boston, MA, USA). Columbia agar base, Mueller-Hinton (MH) agar, and brain heart infusion (BHI) were purchased from Oxiod Ltd. (Basingstoke, Hants, UK). FBS was purchased from Gibco-Life Technologies LLC (Rockville, MD, USA). The sterile defibrinated sheep blood was purchased from Hongquan Biotechnology Co., Ltd. (Guangzhou, Guangdong, China). PrimeSCript™ RT reagent kit with gDNA Eraser and SYBR® Premix Ex Taq™ II (Tli RNaseH Plus) were purchased from Takara Bio Inc. (Kusatsu, Shiga, Japan).

### *H. pylori* strains and growth condition

For the reference *H. pylori* strains, ATCC 43504 strain was obtained from the American Type Culture Collection (ATCC) in Manassas, Virginia, USA, and ATCC 700392 strain was kindly gifted by Professor Hongkai Bi of Nanjing Medical University in China. Sun Yat-sen University (Shenzhen, China) provided the clinical strains CS01, QYZ-001, QYZ-003, and QYZ-004. All clinical strains were identified by the providers via morphological observation, Gram staining, and biochemical reactions. They were then all preserved at -80°C in a solution that contained identical amounts of 65% BHI, 25% glycerol, and 10% FBS [22]. For liquid incubation, bacteria were inoculated in BHI broth supplemented with 10% FBS and shaken at 150 rpm under the same air conditions as stated before [22]. All strains in this study were cultivated at

37˚C for a period of 48 to 72 hours in a tri-gas incubator (BINDER, Germany) providing 10% carbon dioxide (CO2) 5% oxygen (O2), and 85% nitrogen dioxide (N2) [23,24].

## Anti-*H. pylori* activity assays

**Determination of minimum inhibitory concentration (MIC) and minimum bactericidal concentration (MBC).** The MIC assay was conducted using the broth micro-dilution method [24]. In the wells of 96-well microplates, 50 μL of inoculum ($10^6$ CFU/mL as the final dose) and 50 μL of repeated twofold dilutions of CPC in BHI broth were added. The positive control utilized was clarithromycin. The negative control was the culture media containing CPC in BHI broth. Plates were incubated for 3 days at 37˚C in a microaerophilic atmosphere while being shaken at 150 rpm. The MIC was measured by the lack of turbidity in the well and was defined as the minimum concentration that stopped *H. pylori* development. All experiments were performed in triplicate.

Following the measurement of MIC values using the broth microdilution method, the minimum bactericidal concentration (MBC) was established. In a nutshell, 100μl of a solution containing one, two, or four times the MIC concentration of CPC was withdrawn from the 96-well microtiter dish and cultivated for 3 days at 37˚C in a microaerophilic environment. MBC value was determined by a 99.9% decrease in viability as compared to the ones in the control group [22,24]. All experiments were performed in triplicate.

**Inhibiting kinetics assay and killing kinetics assay.** By exposing *H. pylori* to sub-bacteriostatic and bacteriostatic doses of CPC, inhibiting kinetics curves were evaluated. In a nutshell, *H. pylori* ATCC 43504 was given 10% FBS in BHI broth or 1/4MIC, 1/2MIC, or 1 MIC of CPC in BHI broth. Dishes were rotated at 150 rpm for three days at 37˚C in a microaerophilic upper atmosphere. After that, 100 μL of every sample was injected for 600 nm absorbance tests at 0, 8, 12, 24, 28, 32, 36, 48, 60, and 72 h. All experiments were performed in triplicate.

*H. pylori* was subjected to high CPC concentrations to measure the killing kinetics curves. In simple terms, 10% FBS was added to BHI broth before inoculating it with *H. pylori* ATCC 43504. Following administration with the broth from BHI (control) or MBC, 2 MBC, or 4 MBC of CPC for 0, 12, 24, 36, 48, 60, and 72 hours, 100 μL of every sample was collected for several dilutions of tenfold ($1:10$–$1:10^9$). After that, single colonies were formed by plating a 100 μl dilution for 5 days on a Columbia agar basis. The amount of Log10 (CFU/mL) was estimated after the colonies had been counted. All experiments were performed in triplicate.

**Combination with antibiotics: Checkerboard determination.** The *H. pylori* 43504 strain was chosen for the checkerboard analysis of in vitro synergy. To ascertain the synergistic effects of CPC and antibiotics, six successive, two-fold dilutions of CPC and four regularly used medicines (clarithromycin, metronidazole, amoxicillin, and levofloxacin) against *H. pylori* infection were created. Each well in a 96-well plate included 30 μl of CPC, 30μl of antibiotics, and 60μl of *H. pylori* suspensions, resulting in a final bacterial concentration of roughly $1 \times 10^6$ CFU/mL. The plates then underwent incubation at 37˚C in the same air conditions for 3 days while being shaken at 150 rpm. The fractional inhibitory concentration index (FICI) was determined to analyze the relationships between CPC and antibiotics. FICI = $MIC_{CPC(combined)}/MIC_{CPC(alone)} + MIC_{antibiotics(combined)}/MIC_{antibiotics(alone)}$. FICI ≤0.5 was considered synergistic, 0.5 < FICI ≤ 4 was no interaction, and FICI >4 was antagonistic. All experiments were performed in triplicate.

## Analyses of the mechanisms of action on *H. pylori* bacteria

**Scanning electron microscope (SEM).** Scanning electron microscopy (SEM) was used to examine the microscopic structure of *H. pylori* 43504 was adjusted to 1 McFarland (McF)

**Table 1. The order of the study's synthesized primers.**

| Gene | Forward Primer (5′–3′) | Reverse Primer (5′–3′) |
|---|---|---|
| *16S* | CCGCCTACGCGCTCTTTAC | CTAACGAATAAGCACCGGCTAAC |
| *flaA* | ATTGGCGTGTTAGCAGAAGTGA | TGACTGGACCGCCACATC |
| *flaB* | ACATCATTGTGAGCGGTGTGA | GCCCCTAACCGCTCTCAAAT |
| *babA* | TGCTCAGGGCAAGGGAATAA | ATCGTGGTGGTTACGCTTTTG |
| *alpA* | GCACGATCGGTAGCCAGACT | ACACATTCCCCGCATTCAAG |
| *alpB* | ACGCTAAGAAACAGCCCTCAAC | TCATGCGTAACCCCACATCA |
| *ureE* | TCTTGGCTTGGATGTGAATG | GGAATGGTTTGAAACGAGGA |
| *ureF* | GGTCCTGCTGATGGCACTA | GCGTCGTTAGAAGCGTTACG |

before being incubated for 24 hours. Then, 49 mL of BHI broth with 10% FBS either without or with CPC at MIC level was added to 1 mL of the bacterial solution. This was done in a microaerophilic condition shaked at 150 rotations per minute for 12 hours. Bacteria were washed twice with PBS, bacteria were collected by centrifuge at a speed of 6000 rpm for 3 minutes. They were subsequently allowed to cure for 12h at 4°C in 2.5% glutaraldehyde. Before being lyophilized and set the materials received a graded ethanol process of dryness. After metal spraying, the samples had been examined using a Sigma500 scanning electron microscope (ZEISS, Germany).

**RT-qPCR analysis of virulence genes.** To get enough bacteria, 1 McF *H. pylori* ATCC 43504 was cultured for two days. Then, 49 mL of BHI broth containing 10% FBS without or with CPC at MIC concentration was added to 1 mL of the bacterial solution, which was then shaken at 150 rpm for 12 hours in a microaerophilic environment. The germs were gathered and given two PBS washes. The samples' total RNA was extracted as previously described [22]. An American Thermo Scientific NanoDrop 2000 spectrophotometer was used to calculate their amounts. Table 1 contains a list of the appropriate primers used for amplifying mRNA segments. Purified RNA was reverse transcribed using the Takara primeScript RT reagent Kit, and RT-qPCR was performed using the Takara SYBR Premix Ex Tap™ kit per the manufacturer's instructions. A piece of quantitative real-time PCR equipment (7500 Fast Real-Time PCR System, Thermo Scientific, USA) was employed to evaluate the results.

**Measurement of urease enzymatic activity.** The method of urease determination in this study was referred to in the previous study with slight modification [25]. The strains cultured on blood plates for 48h were scraped, the turbidity was adjusted to 1 MCF, and the same volume of 1/2MIC, MIC, and 2MIC CPC and the bacterial solution was added 1:1 in 6-well plates, and set up a growth control group and a positive control group (40 μg/mL and 80 μg/mL of AHA), and then put into the tri-gas incubator for 24 h of incubation at 150 rpm/min. The collected bacterial solution was centrifuged at 6000 rpm/min for 10 min, washed twice with PBS, resuspended in 1 mL of PBS after centrifugation, and adjusted to $OD_{600} = 0.2$. After being diluted with 50 μl of buffer B (25 mM phosphate buffer at pH 6.8, 0.2% Tween-20), the standard suspension of bacterial cells was taken in 50 μl. A 96-well plate with 1 well contained 25 μl of the reduced suspension, which was subsequently diluted with 150 μl of buffer C (25 mM phosphate buffer at pH 6.8, 250 M phenol red), and kept at 37°C for 5 min. A POLARstar Omega (BGM Labtech) plate equipment was used to determine the absorbance at 560 nm every 72 s for 25 rounds after injecting 75 μl of a urea solution (0.5 M) into the wells. The rate of change in absorbance over time was used to quantify activity, which was presented as a proportion of the urease activity of the development of the control sample. The experiment has been carried out at least three times, and all urease activity assessments were made in duplicate.

## Statistical analysis

GraphPad Prism 8.0 software was used to examine the experimental data. The mean values and standard deviation (SD) for the results are provided. One-way ANOVA was used to analyze statistical differences, and Dunnett's multiple comparisons test came next. There was a statistically significant difference, as shown by the $^*P$ value of 0.05.

## Results

### Anti-*H. pylori* activity assays of CPC

**MIC and MBC.** The European Committee on Antimicrobial Susceptibility Testing's (EUCAST 2019) breakpoint values for antibiotic resistance were established. S was for drug-sensitive, R was for drug resistant. Table 2 contains the MIC and MBC values of CPC for two conventional strains of *H. pylori* and four clinically resistant strains. On both clinical and standard strains of *H. pylori*, the CPC showed unique inhibitory actions. The CPC had a MIC of 0.16 to 0.62 μg/mL. MBCs of CPC against multiple H. pylori stains were 0.62 to 1.24 μg/mL and the ratios of MBC/MIC were 2 to 4, which indicated CPC was not only bacteriostatic but also bactericide. In conclusion, the results ascertained that CPC could inhibit a variety of clinically isolated H. pylori strains.

**Time-inhibition curve and time-killing curve.** CPC time-inhibition curves were calculated through the measurement of the OD at 600 nm. The *H. pylori* standard strain ATCC43504 was eliminated by CPC in a time- and dose-dependent fashion (Fig 1A and 1B). The growth of *H. pylori* was significantly impeded when the dose was equal to 1/4MIC. The time-killing curves were created by counting bacterial colonies at various times. As shown in Fig 1B, CPC could kill all *H. pylori* (a 1000-fold decrease in the number of bacteria relative to the initial inoculation) within 12 hours at 2MBCand 4MBC and 24 hours at MBC.

**Combination with antibiotics.** Table 3 displays the interactions between CPC and the four prescription medications that are most frequently used for the treatment of *H. pylori* standard strain ATCC43504 infection. Even though CPC had no synergistic effect with any of the antibiotics tested in this investigation, CPC had no interaction effects. According to these results, there was no antagonism between CPC and the four antibiotics in combination, suggesting that CPC has the potential to be combined with all four antibiotics.

### The mechanisms of action of CPC against *H. pylori*

**Effect of CPC on the morphology of *H. pylori*.** SEM was used to examine the morphology of *H. pylori* with and without CPC during MIC concentration incubation (Fig 2). In the CPC treatment groups, the bacterial cells' surfaces shrank and the membrane was broken; also, the CPC group may lead to spherical form in bacteria. SEM showed that the surface of control cells lacking CPC was soft and consistent, with a slightly curved rod-shaped shape.

**Table 2. The results of CPC's MIC and MBC measurements on several *H. pylori* strain.**

| *H. pylori* strains | Drug sensitivity | MIC(μg/mL) | MBC(ug/mL) | MBC/MIC |
|---|---|---|---|---|
| ATCC43504 | R(MTZ) | 0.31 | 0.62 | 2 |
| ATCC700392 | S | 0.31 | 0.62 | 2 |
| CS01 | R(CLR) | 0.16 | 0.31 | 2 |
| QYZ-001 | R(MTZ) | 0.62 | 1.24 | 2 |
| QYZ-003 | R(MTZ, CLR, LEF) | 0.31 | 1.24 | 4 |
| QYZ-004 | R(MTZ, CLR, LEF, AMO) | 0.31 | 0.62 | 2 |

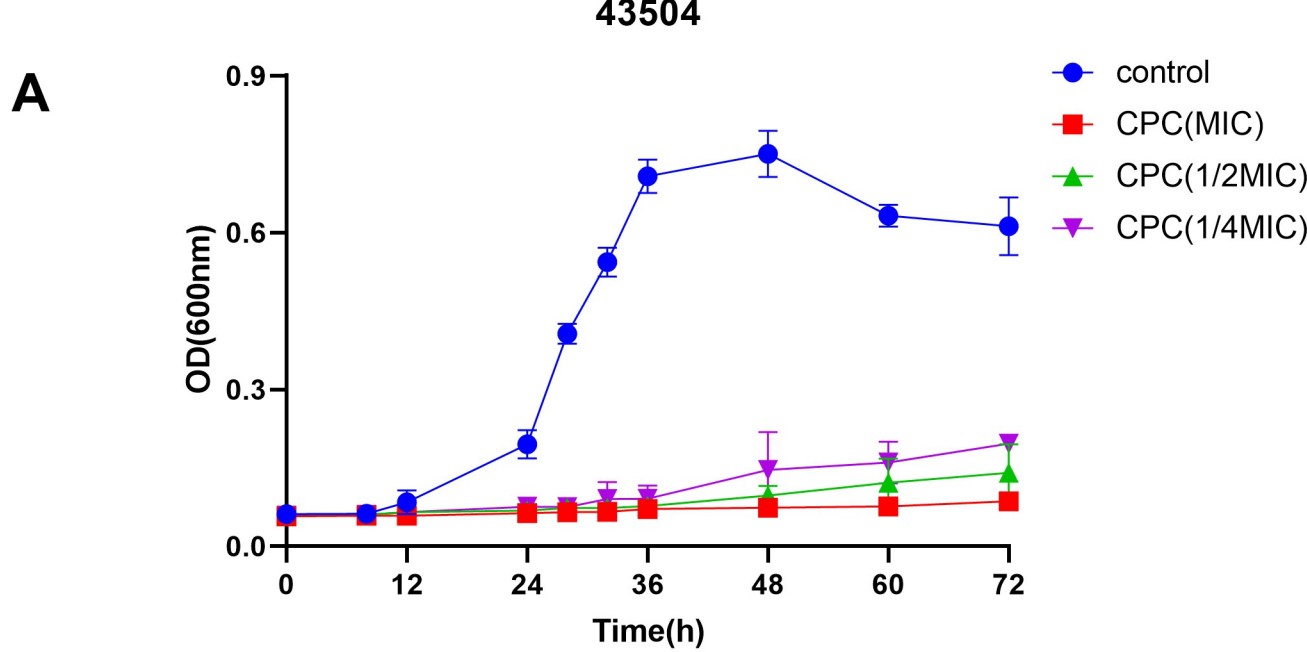

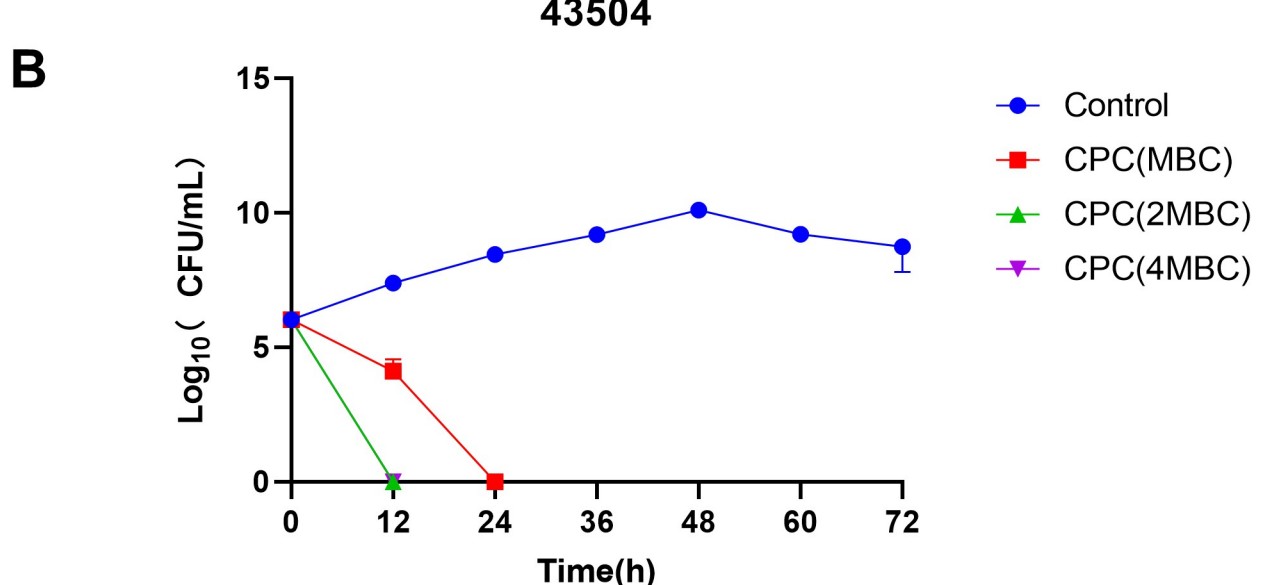

**Fig 1. In vitro anti-*H. pylori* activity of CPC.** (A) Inhibiting kinetics curves of CPC on ATCC 43504. (B) Killing kinetics curves of CPC on ATCC 43504.

**Effects of CPC on the expression of *H. pylori's* virulence genes.** Based on the results shown in Fig 3, it is evident that CPC (presumably referring to a specific compound or treatment) caused a significant decrease in the expression of *flaA*, *flaB*, *babA*, *alpA*, *alpB*, *ureE*, and *ureF* in ATCC43504 bacteria. This indicates that CPC has a down-regulatory effect on the mentioned genes in this bacterial strain.

**Table 3. MICs of CPC were used either alone or in conjunction with antibiotics.**

| Combination | MIC(μg/mL) | | | FICI | Interaction |
|---|---|---|---|---|---|
| | CPC | Antibiotic | CPC+Antibiotic | | |
| CPC+CLR | 0.31 | 0.016 | 0.31+0.016 | 2 | no interaction |
| CPC+MTZ | 0.31 | 0.2 | 0.16+0.2 | 1.5 | no interaction |
| CPC+LEF | 0.31 | 0.4 | 0.31+0.2 | 1.5 | no interaction |
| CPC+AMO | 0.31 | 1 | 0.31+0.25 | 1.25 | no interaction |

**Impact of CPC on the urease activity of *H. pylori*.** The change in urease activity of *H. pylori* with drug concentration is shown in Fig 4. Urease is one of the key factors for the successful colonization of *H. pylori* in the stomach [26–30]. Under high acid conditions, *H. pylori* can produce a large amount of urease. This urease can hydrolyze urea into ammonia, which can neutralize gastric acid and create an environment suitable for the growth of *H. pylori* [25,31,32]. As a result, the goal of decreasing urease activity is to stop the proliferation of *H. pylori*. As shown in Fig 4, the urease inhibitor AHA can significantly inhibit urease activity at both 40 μg/mL and 80 μg/mL concentrations for ATCC 43504 strain. There was a significant inhibition of urease activity after *H. pylori* dosing after 24h of CPC given at 1/2 MIC compared to the control group. And after administration of more than MIC, an inhibitory effect equivalent to that of the positive drug was achieved.

## Discussion

During the MIC and MBC determination, we observed strong inhibitory and bactericidal effects of CPC on various *H. pylori* strains, including both antibiotic-susceptible and -resistant strains. The MBC/MIC values were 2 as well as 4, while the MIC values (100 percent inhibition of bacterial growth) varied from 0.16 ug/mL to 0.62 ug/mL. MIC values of classical antibiotics ranged from 0.016 μg/mL to 1μg/mL (Table 3). Studies reporting similar results for killing of *H. pylori* bacteria by classical antibiotics [22–24]. Although CPC has demonstrated substantial toxicity in high concentrations, it is regarded as safe at quantities below 0.1% [14,33,34]. Our study shows that the effective concentration of CPC is well below 0.1%. It was found that CPC showed a certain level of bacterial inhibition at 0.16 ug/mL and that it exhibited bactericidal effects after just 12 hours of exposure during the research of inhibitory kinetics and killing kinetics. The comparable potency of CPC to classical antibiotics as observed in its bactericidal effects after only 12 hours of exposure indicates its potential as a promising agent for *H. pylori* treatment. In addition, currently marketed products include the CPC buccal tablets, which are effective in increasing the residence time of the drug in the stomach, providing a time advantage in treating *H. pylori*, and it has also proven to be a safe drug.

Drug resistance in *H. pylori* has been studied based on molecular [35] and genetic levels [36] but has not been effectively addressed. To assist prevent the spread of antibiotic resistance, we therefore also explored the option of supplementing CPC with antibiotics for the therapy of *H. pylori*. The interaction between CPC and four widely used antibiotics was examined using the checkerboard dilution method. Our findings demonstrated that CPC wasn't interacting synergistically with any of the antibiotics evaluated, but rather had neutral effects, suggesting that they could be administered alongside these drugs without having an adverse reaction.

It has been found that the mechanism of killing *H. pylori* is to cause changes in the visual morphology [37] of the bacterium and the integrity of the bacterial cell membrane [38–40]. CPC is an amphiphilic chemical that disrupts cytoplasmic membranes in bacteria as part of its recognized mechanism of action [16]. As was previously demonstrated for the influenza virus

## Control    CPC-MIC

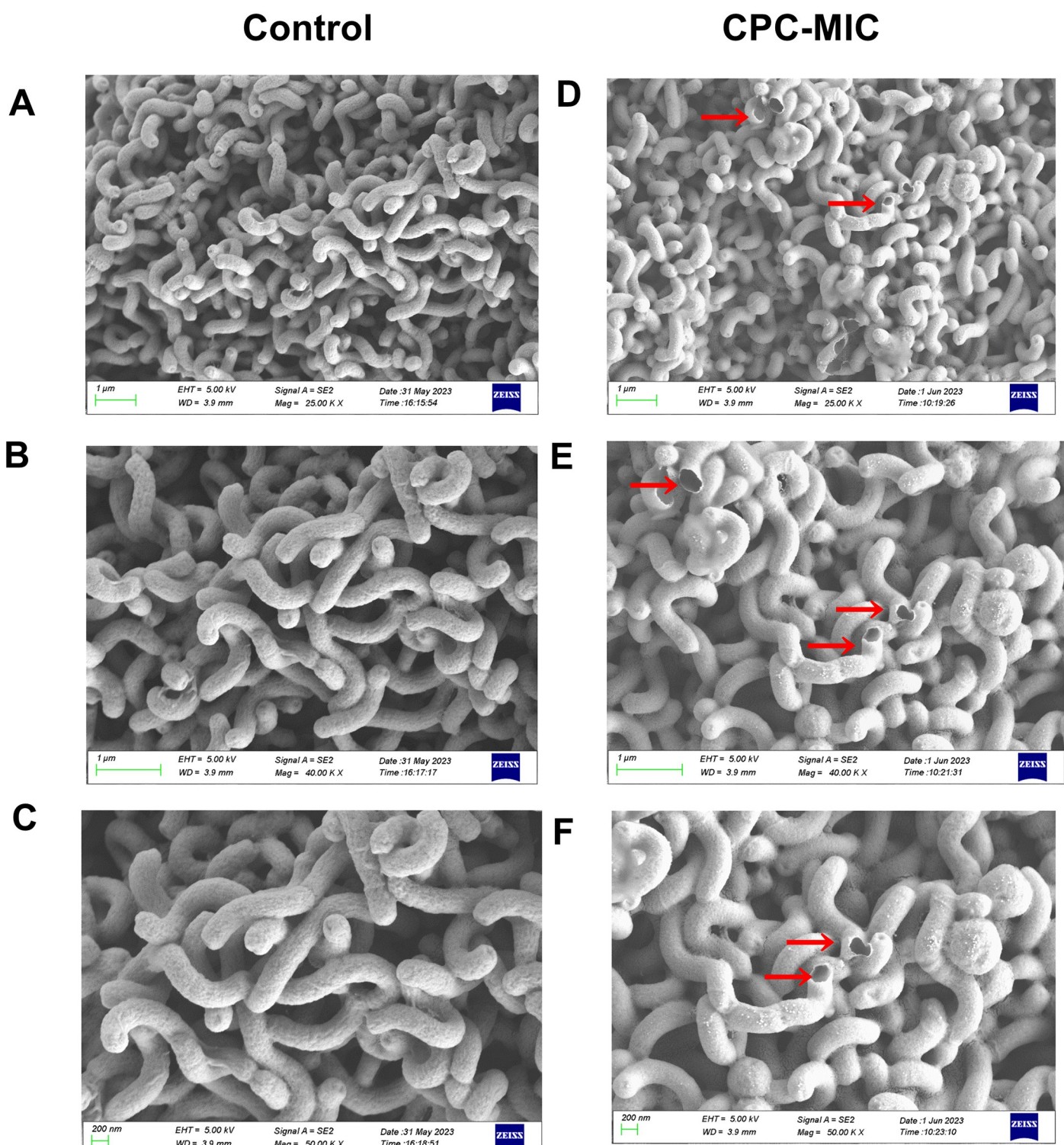

**Fig 2. The SEM images of *H. pylori* 43504 after various treatments.** Morphological pictures of *H. pylori* cells on SEM (magnifications of 25.0 kx, 40 kx, and 50.0 kx) of control (A, B, and C) and CPC treatment (D, E, and F) at MIC dose for 12 hours. The membrane damage is highlighted by the red arrows.

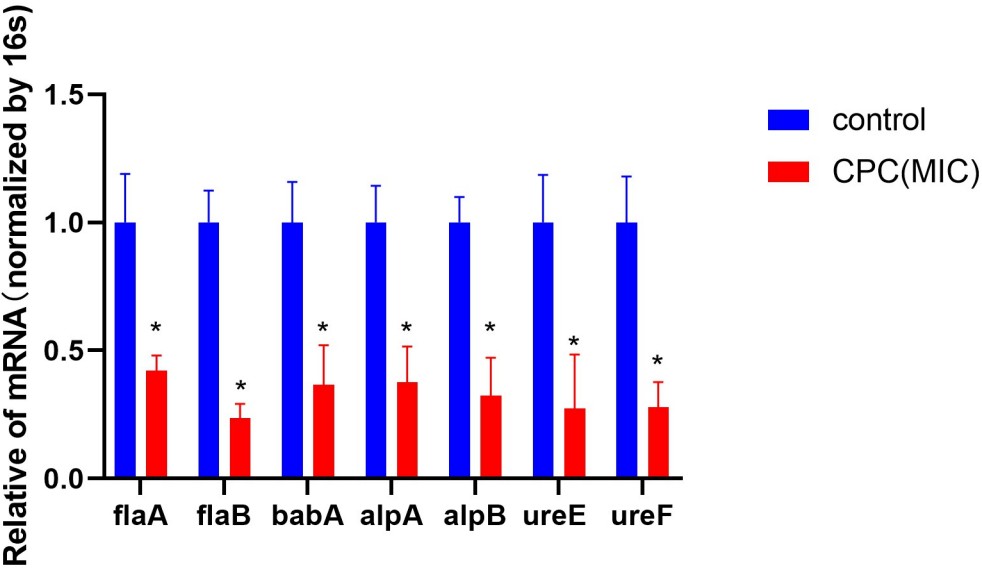

**Fig 3. RT-qPCR analysis results showed the effects of CPC on the mRNA levels of *H. pylori* virulence genes.** The bacteria strain was ATCC 43504, and the concentration of CPC was 0.31 μg/mL (MIC). And the incubation time of CPC was 12 h *$P < 0.05$, vs control group.

[20], the virucidal effect of CPC in enclosed viruses is similarly mediated by the destruction of the lipid bilayer, and it also affects SARS-CoV-2 [21]. After just 12 hours of treatment at MIC, the SEM findings for morphological observations suggested that CPC significantly affected the morphology of the *H. pylori* cell membrane and interfered with the integrity, which may have caused bleb formation and cell lysis [22–24].

The pathogenesis of *H. pylori* infection can be related to a multifaceted interaction of host, surroundings, and bacterial virulence variables [41]. The urea-transporter genes *flaA*, *flaB*,

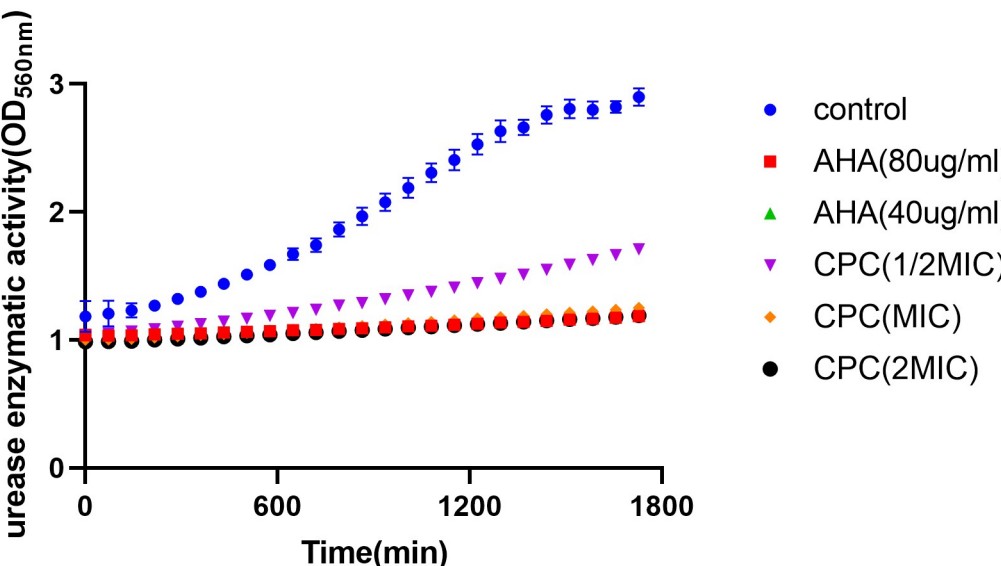

**Fig 4. The inhibition of urease enzymatic activity induced by CPC.** The concentrations of CPC were 1/2MIC, MIC, and 2MIC, while the concentration range of AHA (positive control) was 40 and 80 μg/mL.

*babA*, *alpA*, *alpB*, *ureE*, and *ureF* in ATCC43504 bacteria were downregulated by CPC in the current study. The movement of *H. pylori* is crucial for its successful colonization of the gastric epithelium, and this is facilitated by the motility of *H. pylori*. [42,43]. *H. pylori* possesses lophotrichous flagella, which are composed of flagellin proteins *flaA* and *flaB*, and these proteins form the filament structure of the flagella. [44–46]. Deletion of the *flaA* and *flaB* genes in *H. pylori* has been reported to result in a lack of motility [47], and down-regulation of the flagellum-associated gene *flaA* by hesperetin may also contribute to the inhibition of colonization [46]. It can be implied that the inhibition of *H. pylori* colonization may involve the downregulation of flagella components by CPC. Hesperetin inhibited the expression of *alpA*, *alpB*, and *babA* genes, which are known as *H. pylori* adherence-related genes [48–50]. The adherence ability of *H. pylori* has been suggested as a mechanism for chronic gastritis and gastric cancer induced by *H. pylori* [51]. The results imply that reducing long-term infection and inflammatory response of *H. pylori* can be achieved by inhibiting *H. pylori* adhesion using CPC. Long-term *H. pylori* colonization in the extremely acidic intestinal environment is heavily reliant on urease [52]. And the consequences of *H. pylori* urease's ammonia production are direct damage to gastric epithelial cells and the induction of an immune response [53]. To combat *H. pylori* infections, extensive research has been carried out on drugs that specifically target urease, such as propolis [54], Zanthoxylum nitidum [55], and Canarium album Raeusch. fruit extracts [24]. *UreE* and *ureF* gene expression inhibition by CPC can be observed in Fig 3, while urease activity inhibition can be observed in Fig 4. These findings show that inhibiting *H. pylori ureE* and *ureF* genes with CPC can lower urease production, which is consistent with the findings of the current research.

## Conclusion

The current investigation demonstrated their significant impact on both antibiotic-sensitive and antibiotic-resistant *H. pylori* strains. CPC was also a bactericide for each of the strains evaluated. Furthermore, CPC's mechanisms of action against *H. pylori* have been identified as bacterial structural destruction, down-regulation of virulence gene expression, and reduced urease activity. This research could serve as a reference for upcoming anti-*H. pylori* studies or clinical CPC applications.

## Supporting information

**S1 File.**
(DOCX)

## Acknowledgments

We thank for Chang Peng, Bingmei Su, Yuqian Lai, Meiyun Chen and Liping Zhang for the help of the data extraction and data analysis.

## Author Contributions

**Conceptualization:** Mingjin Xun, Meicun Yao.

**Data curation:** Mingjin Xun, Ruixia Wei, Zimao Fan.

**Formal analysis:** Mingjin Xun, Ruixia Wei, Zimao Fan.

**Funding acquisition:** Guimin Zhang.

**Methodology:** Zimao Fan.

**Resources:** Haibo Wang.

**Software:** Haibo Wang.

**Supervision:** Haibo Wang.

**Validation:** Haibo Wang.

**Writing – original draft:** Mingjin Xun, Zhong Feng, Hui Li, Junwei Jia, Xiaoyan Shi, Zhanzhu Lv.

**Writing – review & editing:** Mingjin Xun, Zhong Feng, Hui Li, Meicun Yao, Junwei Jia, Xiaoyan Shi, Zhanzhu Lv.

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
