## [Decision Letter · Decision Letter 0]

23 Jan 2024

PONE-D-23-40407In vitro anti-Helicobacter pylori activity and preliminary mechanism of action of Cetylpyridinium ChloridePLOS ONE Dear Dr. xun,

Thank you for submitting your manuscript to PLOS ONE. After careful consideration, we feel that it has merit but does not fully meet PLOS ONE’s publication criteria as it currently stands. Therefore, we invite you to submit a revised version of the manuscript that addresses the points raised during the review process.

We look forward to receiving your revised manuscript.

Kind regards,

Samiullah Khan, Ph. D

Academic Editor

PLOS ONE

3. Please amend the manuscript submission data (via Edit Submission) to include authors Mingcun Yao and Haibo Wang.

4. Please amend your authorship list in your manuscript file to include author meicun Yao,.

Additional Editor Comments:

Dear author,

Revise the whole manuscript thoroughly in the light of comments/suggestions of all the three reviewers and submitt it again.

Reviewers' comments:

Reviewer's Responses to Questions

**Comments to the Author**

1. Is the manuscript technically sound, and do the data support the conclusions?

Reviewer #1: Yes

Reviewer #2: Yes

Reviewer #3: Partly

2. Has the statistical analysis been performed appropriately and rigorously? 

Reviewer #1: Yes

Reviewer #2: Yes

Reviewer #3: Yes

3. Have the authors made all data underlying the findings in their manuscript fully available?

Reviewer #1: Yes

Reviewer #2: Yes

Reviewer #3: Yes

4. Is the manuscript presented in an intelligible fashion and written in standard English?

Reviewer #1: Yes

Reviewer #2: Yes

Reviewer #3: Yes

5. Review Comments to the Author

Reviewer #1: After reading the original article entitled "In vitro anti-Helicobacter pylori activity and preliminary mechanism of action of Cetylpyridinium Chloride", I believe that the manuscript is interesting and contains promising results with MIC values of compounds at the activity level of classically used antibiotics.

I only have a few minor comments that I would like the Authors to address:

- In general, it seems to me that the phrase "mechanism of activity" in the context of antimicrobial substances is rather reserved for cellular components, e.g. cell membrane, cell wall, genetic material or ribosomes; since such studies were not performed and the focus was on the expression of virulence factors, I would suggest describing it (in the title and text) as "antivirulence activity"

- page 3: "copper treatment" - I think it should be bismuth

- page 4: "as well as SARS-CoV-2" -> including SARS-CoV-2 (this virus is also enveloped)

- Classically, in the FICI classification, values >4 are considered antagonistic

- In the "Results" section: why were the EUCAST 2019 recommendations followed and not the newer ones, e.g. from 2023?

- In the "Conclusions" section, remove the sentence "It can also be taken with clarithromycin, metronidazole, levofloxacin and amoxicillin"

Reviewer #2: 1) Cetylpyridinium chloride...should be small letter in the tittle & keywords

2) mechanism of action, In vitro - please use small letter in keywords

3) Please standardized Fig. & Figure

4) Intodution - last part...This research could serve.....should be in conclusion part.

5)Conclusion - IT can be taken with clarithromycin... i think this should be further investigated

Reviewer #3: The authors designed an experiment to find the synergistic effect of CPC and different antibiotics. They found that the effect is additive and not synergistic. According to the literature, CPC is known as disinfectant and it has known effect on the lipid membranes. Also, it is predicted that the disruption of the cell membrane will decrease cell metabolism and other vital functions. The experiment on inhibiting urease enzyme is not enough and other experiments for the binding of CPC to the enzyme, if it does not work as non-specific inhibitor. Accordingly, I would not recommend publishing this manuscript in the current form. It needs further experimentation to confirm the proposed mechanism of action of CPC.

1. For (FICI), please reference this.

2. “The CPC had a MIC of 0.16 to 0.62 μg/ml, while the MBC/MIC had a 2-4, which was thought to have antibacterial properties.”

Please mention the reference for such values to decide it has antibiotic properties.

3. Table 2 is not clear.

You said the paragraph “Table 2 contains the MIC and MBC values of CPC for two conventional strains of H. pylori and four clinically resistant strains. On both clinical and standard strains of H. pylori, the CPC showed unique inhibitory actions.”

And nothing in the table explains the CPC activity. This table needs to be more clear, or explain it in the paragraph.

4. “According to these results, CPC could be given along with the other four antibiotics without having an adverse effect.”

What do you mean by this? Please explain this in more details.

5. Figure 2, no red arrows present.

6. According to figure 3, mRNA is for all virulence genes you mentioned in the experimental part and specifically for each one. I suppose that you will find this in any other mRNA for any transcript. I would suggest doing this amplification and measuring the amount of mRNA of other mRNA (no-virulence genes).

6. PLOS authors have the option to publish the peer review history of their article (what does this mean?). If published, this will include your full peer review and any attached files.

Reviewer #1: No

Reviewer #2: No

Reviewer #3: No

---

## [Author Response · Author response to Decision Letter 0]

14 Feb 2024

Dear editors and reviewers,

We would like to thank you and the reviewers for the constructive comments and suggestions. According with your advice, we amended the relevant part in manuscript. All of your questions were answered below.

Reviewer #1

Recommendation 1. In general, it seems to me that the phrase "mechanism of activity" in the context of antimicrobial substances is rather reserved for cellular components, e.g. cell membrane, cell wall, genetic material or ribosomes; since such studies were not performed and the focus was on the expression of virulence factors, I would suggest describing it (in the title and text) as "antivirulence activity".

Response: Thank you for your recommendation. We have revised the title to “In vitro anti-Helicobacter pylori activity and antivirulence activity of cetylpyridinium chloride”. 

Recommendation 2. page 3: "copper treatment" - I think it should be bismuth

Response: Thanks for your recommendation. We are very sorry for our incorrect writing. We have revised it in manuscript.

Recommendation 3. page 4: "as well as SARS-CoV-2" -> including SARS-CoV-2 (this virus is also enveloped)

Response: Thank you for your recommendation. We have revised “as well as” to “including” in the manuscript.

Recommendation 4. Classically, in the FICI classification, values >4 are considered antagonistic

Response: Thank you for your recommendation. For the FICI study, we did it with reference to the literature [1,2]. After your suggestion, we have re-investigated some of the literature [3,4]. Since there is a widely accepted criteria in MIC testing, that variation in a single result places an MIC in a three-dilution range (mode ± 1 dilution), the possibilities for reproducibility errors in an MIC chequerboard are considerable. As a result, Value >4 Considered antagonistic are more scientific. Thanks again for your suggestions, which we've also revised in our manuscript.

References

[1] Peng, C., Sang, S., Shen, X., Zhang, W., Yan, J., Chen, P., Jiang, C., Yuan, Y., Zhu, W., Yao, M., 2022. In vitro anti-helicobacter pylori activity of syzygium aromaticum and the preliminary mechanism of action. J. Ethnopharmacol. 288, 114995. https://doi.org/10.1016/j.jep.2022.114995.

[2] Yan, J., Peng, C., Chen, P., Zhang, W., Jiang, C., Sang, S., Zhu, W., Yuan, Y., Hong, Y., Yao, M., 2022. In-vitro anti-helicobacter pylori activity and preliminary mechanism of action of canarium album raeusch. Fruit extracts. J. Ethnopharmacol. 283, 114578. https://doi.org/10.1016/j.jep.2021.114578.

[3] Shen, X., Zhang, W., Peng, C., Yan, J., Chen, P., Jiang, C., Yuan, Y., Chen, D., Zhu, W., Yao, M., 

2021. In vitro anti‐bacterial activity and network pharmacology analysis of sanguisorba officinalis l. Against helicobacter pylori infection. Chin. Med. 16 (1). https://doi.org/10.1186/s13020-021-00442-1.

[4] Odds, F.C., 2003. Synergy, antagonism, and what the chequerboard puts between them. J. Antimicrob. Chemother. 52 (1), 1.

Recommendation 5. In the "Results" section: why were the EUCAST 2019 recommendations followed and not the newer ones, e.g. from 2023?

Response: Thank you for your comments. For the EUCAST 2019 recommendations, we referenced the team's previous research methods [1,2,3]. In the meantime, we checked and found that this is the latest version (eucast: New S, I and R definitions). 

References

[1] Peng, C., Sang, S., Shen, X., Zhang, W., Yan, J., Chen, P., Jiang, C., Yuan, Y., Zhu, W., Yao, M., 2022. In vitro anti-helicobacter pylori activity of syzygium aromaticum and the preliminary mechanism of action. J. Ethnopharmacol. 288, 114995. https://doi.org/10.1016/j.jep.2022.114995.

[2] Yan, J., Peng, C., Chen, P., Zhang, W., Jiang, C., Sang, S., Zhu, W., Yuan, Y., Hong, Y., Yao, M., 2022. In-vitro anti-helicobacter pylori activity and preliminary mechanism of action of canarium album raeusch. Fruit extracts. J. Ethnopharmacol. 283, 114578. https://doi.org/10.1016/j.jep.2021.114578.

[3] Shen, X., Zhang, W., Peng, C., Yan, J., Chen, P., Jiang, C., Yuan, Y., Chen, D., Zhu, W., Yao, M., 

2021. In vitro anti‐bacterial activity and network pharmacology analysis of sanguisorba officinalis l. Against helicobacter pylori infection. Chin. Med. 16 (1). https://doi.org/10.1186/s13020-021-00442-1.

Recommendation 6. In the "Conclusions" section, remove the sentence "It can also be taken with clarithromycin, metronidazole, levofloxacin and amoxicillin"

Response: Thank you for your recommendation. We have removed this sentence from the conclusion.

Reviewer #2: 

Recommendation 1. Cetylpyridinium chloride...should be small letter in the tittle & keywords

Response: Thanks for your recommendation. We are very sorry for our incorrect writing. We have revised it in the manuscript.

Recommendation 2. mechanism of action, In vitro - please use small letter in keywords

Response: Thanks for your recommendation. We are very sorry for our incorrect writing. We have revised it in the manuscript.

Recommendation 3. Please standardized Fig. & Figure

Response: Thank you for your recommendation. We have revised all “Figure “ to “Fig.”.

Recommendation 4. Introduction - last part...This research could serve.....should be in conclusion part.

Response: Thank you for your recommendation. We have deleted the sentence “This research could serve.....” in the manuscript.

Recommendation 5. Conclusion - IT can be taken with clarithromycin... i think this should be further investigated

Response: Thank you for your comments. We have deleted this sentence from the conclusion. In a follow-up study we will use animal experiments to investigate this issue in depth.

Reviewer #3: 

Recommendation 1. For (FICI), please reference this.

Response: Thank you for your recommendation, and I am so sorry for not fully understood your comments “please reference this”, if the following response doesn’t address what you recommended, please let me know and we will response once again as soon as possible.

For the FICI study, we did it with reference to the literature [1,2]. After your suggestion, we have re-investigated some of the literature [3,4]. Since there is a widely accepted criteria in MIC testing, that variation in a single result places an MIC in a three-dilution range (mode ± 1 dilution), the possibilities for reproducibility errors in an MIC chequerboard are considerable. As a result, Value >4 Considered antagonistic are more scientific, which we've also revised in our manuscript.

References

[1] Peng, C., Sang, S., Shen, X., Zhang, W., Yan, J., Chen, P., Jiang, C., Yuan, Y., Zhu, W., Yao, M., 2022. In vitro anti-helicobacter pylori activity of syzygium aromaticum and the preliminary mechanism of action. J. Ethnopharmacol. 288, 114995. https://doi.org/10.1016/j.jep.2022.114995.

[2] Yan, J., Peng, C., Chen, P., Zhang, W., Jiang, C., Sang, S., Zhu, W., Yuan, Y., Hong, Y., Yao, M., 2022. In-vitro anti-helicobacter pylori activity and preliminary mechanism of action of canarium album raeusch. Fruit extracts. J. Ethnopharmacol. 283, 114578. https://doi.org/10.1016/j.jep.2021.114578.

[3] Shen, X., Zhang, W., Peng, C., Yan, J., Chen, P., Jiang, C., Yuan, Y., Chen, D., Zhu, W., Yao, M., 

2021. In vitro anti‐bacterial activity and network pharmacology analysis of sanguisorba officinalis l. Against helicobacter pylori infection. Chin. Med. 16 (1). https://doi.org/10.1186/s13020-021-00442-1.

[4] Odds, F.C., 2003. Synergy, antagonism, and what the chequerboard puts between them. J. Antimicrob. Chemother. 52 (1), 1.

Recommendation 2. “The CPC had a MIC of 0.16 to 0.62 μg/ml, while the MBC/MIC had a 2-4, which was thought to have antibacterial properties.” Please mention the reference for such values to decide it has antibiotic properties.

Response: Thanks for your recommendation. We have revised it in the manuscript. Please see the updates below.

The CPC had a MIC of 0.16 to 0.62 μg/ml. MBCs of CPC against multiple H. pylori stains were 0.62 to 1.24 µg/ml and the ratios of MBC/MIC were 2 to 4, which indicated CPC was not only bacteriostatic but also bactericide. In conclusion, the results confirm that CPC could inhibit a variety of clinically isolated H. pylori strains. 

Our team has done the inhibitory actions of antibiotics and the MIC is 0.004-1.6μg/ml [1,2], however the MIC for herbal medicine is 80-640[1,2,3]. For CPC, the MIC is 0.16-0.62μg/ml, which is a very low concentration.

References.

[1] Peng, C., Sang, S., Shen, X., Zhang, W., Yan, J., Chen, P., Jiang, C., Yuan, Y., Zhu, W., Yao, M., 2022. In vitro anti-helicobacter pylori activity of syzygium aromaticum and the preliminary mechanism of action. J. Ethnopharmacol. 288, 114995. https://doi.org/10.1016/j.jep.2022.114995.

[2] Yan, J., Peng, C., Chen, P., Zhang, W., Jiang, C., Sang, S., Zhu, W., Yuan, Y., Hong, Y., Yao, M., 2022. In-vitro anti-helicobacter pylori activity and preliminary mechanism of action of canarium album raeusch. Fruit extracts. J. Ethnopharmacol. 283, 114578. https://doi.org/10.1016/j.jep.2021.114578.

[3] Shen, X., Zhang, W., Peng, C., Yan, J., Chen, P., Jiang, C., Yuan, Y., Chen, D., Zhu, W., Yao, M., 

2021. In vitro anti‐bacterial activity and network pharmacology analysis of sanguisorba officinalis l. Against helicobacter pylori infection. Chin. Med. 16 (1). https://doi.org/10.1186/s13020-021-00442-1.

Recommendation 3. Table 2 is not clear. You said the paragraph “Table 2 contains the MIC and MBC values of CPC for two conventional strains of H. pylori and four clinically resistant strains. On both clinical and standard strains of H. pylori, the CPC showed unique inhibitory actions.” And nothing in the table explains the CPC activity. This table needs to be more clear, or explain it in the paragraph.

Response: Thanks for your recommendation. Minimal inhibitory concentration (MIC) and minimum bactericidal concentration (MBC) are concentration of a drug that minimizes the proliferation or kills the survival of pathogenic microorganisms and are used to measure the ability of anti-infective drugs (including antibiotics, antimicrobials, and synthetic chemicals) to resist pathogenic microorganisms, so we think the MIC and MBC values of CPC can represent the CPC activity.

Recommendation 4. “According to these results, CPC could be given along with the other four antibiotics without having an adverse effect.” What do you mean by this? Please explain this in more details.

Response: Thank you for your comments. We are very sorry for our incorrect writing. We have revised the manuscript. There was no antagonism between CPC and the four antibiotics in combination, suggesting that CPC has the potential to be combined with all four antibiotics.

Recommendation 5. Figure 2, no red arrows present.

Response: Thank you for your comments. We have revised it in the manuscript.

Recommendation 6. According to figure 3, mRNA is for all virulence genes you mentioned in the experimental part and specifically for each one. I suppose that you will find this in any other mRNA for any transcript. I would suggest doing this amplification and measuring the amount of mRNA of other mRNA (no-virulence genes).

Response: Thank you for your comments. For this study, we choose virulence genes by referencing our team's previous research experience and other references [1,2,3]. For no-virulence genes, we amplified and measured the 16s as the Reference genes. Our team has not paid enough attention at present and we will increase this kind of research work in the subsequent experiments.

References

[1] Peng, C., Sang, S., Shen, X., Zhang, W., Yan, J., Chen, P., Jiang, C., Yuan, Y., Zhu, W., Yao, M., 2022. In vitro anti-helicobacter pylori activity of syzygium aromaticum and the preliminary mechanism of action. J. Ethnopharmacol. 288, 114995. https://doi.org/10.1016/j.jep.2022.114995.

[2] Yan, J., Peng, C., Chen, P., Zhang, W., Jiang, C., Sang, S., Zhu, W., Yuan, Y., Hong, Y., Yao, M., 2022. In-vitro anti-helicobacter pylori activity and preliminary mechanism of action of canarium album raeusch. Fruit extracts. J. Ethnopharmacol. 283, 114578. https://doi.org/10.1016/j.jep.2021.114578.

[3] Ou, L., Liu, H. R., Shi, X. Y., Peng, C., Zou, Y. J., Jia, J. W., Yao, M., 2024. Terminalia chebula Retz. aqueous extract inhibits the Helicobacter pylori-induced inflammatory response by regulating the inflammasome signaling and ER-stress pathway. J.Ethnopharmacology, 320, 117428.

Thank you and all the reviewers again for the kind advice. Should you have any questions, please contact us without any hesitate.

Sincerely yours,

Guimin Zhang, Zhong Feng

---

## [Decision Letter · Decision Letter 1]

5 Mar 2024

In vitro anti-Helicobacter pylori activity and antivirulence activity of cetylpyridinium chloride

PONE-D-23-40407R1

Dear Dr.mingjin xun ,

We’re pleased to inform you that your manuscript has been judged scientifically suitable for publication and will be formally accepted for publication once it meets all outstanding technical requirements.

Kind regards,

Samiullah Khan, Ph. D

Academic Editor

PLOS ONE

Additional Editor Comments (optional):

Reviewers' comments:

Reviewer's Responses to Questions

**Comments to the Author**

1. If the authors have adequately addressed your comments raised in a previous round of review and you feel that this manuscript is now acceptable for publication, you may indicate that here to bypass the “Comments to the Author” section, enter your conflict of interest statement in the “Confidential to Editor” section, and submit your "Accept" recommendation.

Reviewer #1: All comments have been addressed

2. Is the manuscript technically sound, and do the data support the conclusions?

Reviewer #1: Yes

3. Has the statistical analysis been performed appropriately and rigorously? 

Reviewer #1: Yes

4. Have the authors made all data underlying the findings in their manuscript fully available?

Reviewer #1: Yes

5. Is the manuscript presented in an intelligible fashion and written in standard English?

Reviewer #1: Yes

6. Review Comments to the Author

Reviewer #1: Authors have made all the corrections needed. I believe that the manuscript is suitable for publishing.

7. PLOS authors have the option to publish the peer review history of their article (what does this mean?). If published, this will include your full peer review and any attached files.

Reviewer #1: No

---

## [Editor Report · Acceptance letter]

1 Apr 2024

PONE-D-23-40407R1 

PLOS ONE

Dear Dr. xun, 

I'm pleased to inform you that your manuscript has been deemed suitable for publication in PLOS ONE. Congratulations! Your manuscript is now being handed over to our production team.

Kind regards, 

on behalf of

Dr. Samiullah Khan 

Academic Editor

PLOS ONE